# UNER: A Unified Prediction Head for Named Entity Recognition in Visually-rich Documents

## ABSTRACT

The recognition of named entities in visually-rich documents (VrD-NER) plays a critical role in various real-world scenarios and applications. However, the research in VrD-NER faces three major challenges: complex document layouts, incorrect reading orders, and unsuitable task formulations. To address these challenges, we propose a query-aware entity extraction head, namely UNER, to collaborate with existing multi-modal document transformers to develop more robust VrD-NER models. The UNER head considers the VrD-NER task as a combination of sequence labeling and reading order prediction, effectively addressing the issues of discontinuous entities in documents. Experimental evaluations on diverse datasets demonstrate the effectiveness of UNER in improving entity extraction performance. Moreover, the UNER head enables a supervised pre-training stage on various VrD-NER datasets to enhance the document transformer backbones and exhibits substantial knowledge transfer from the pre-training stage to the fine-tuning stage. By incorporating universal layout understanding, a pre-trained UNER-based model demonstrates significant advantages in few-shot and cross-linguistic scenarios and exhibits zero-shot entity extraction abilities.

## CCS CONCEPTS

• **Computing methodologies** → **Information extraction**; Transfer learning; • **Applied computing** → *Document analysis*.

## KEYWORDS

UNER; Named Entity Recognition; Document Understanding; Supervised Pre-training

## 1 INTRODUCTION

Named Entity Recognition (NER) on Visually-rich Documents (VrDs), referred to as VrD-NER, is a task that aims to identify user-specified entities in diverse types of documents. This research topic holds significant importance due to its wide applicability in real-world scenarios. In recent years, the VrD-NER task has witnessed advancements through the utilization of pre-training techniques in NLP [2, 4] and CV [1, 12]. Many multi-modal document transformers have emerged, such as the LayoutLM model series [6, 30, 33]. Typically, these document transformers are first pre-trained on a massive document corpus with self-supervised pre-training tasks

Permission to make digital or hard copies of all or part of this work for personal or classroom use is granted without fee provided that copies are not made or distributed for profit or commercial advantage and that copies bear this notice and the full citation on the first page. Copyrights for components of this work owned by others than the author(s) must be honored. Abstracting with credit is permitted. To copy otherwise, or republish, to post on servers or to redistribute to lists, requires prior specific permission and/or a fee. Request permissions from permissions@acm.org.

*ACM MM, 2024, Melbourne, Australia*

© 2024 Copyright held by the owner/author(s). Publication rights licensed to ACM.
ACM ISBN 978-x-xxxx-xxxx-x/YY/MM
https://doi.org/10.1145/nnnnnnn.nnnnnnn

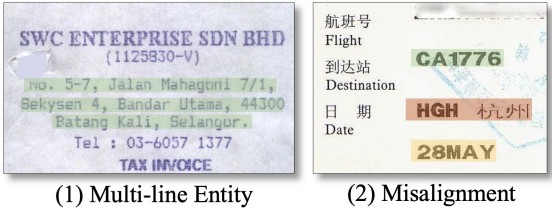

(1) Multi-line Entity      (2) Misalignment

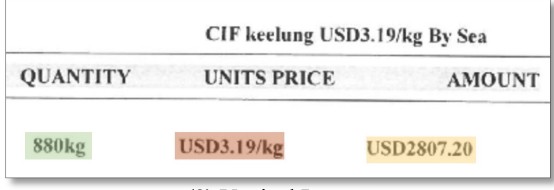

(3) Vertical Layout

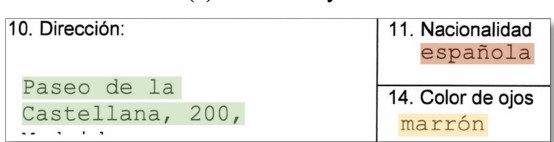

(4) Irregular Layout

**Figure 1: Illustration of the common issues in the VrD-NER problem. We utilize color-coded words to indicate entities and words of the same color signify a complete entity span. These issues contribute to the complexity of understanding the document and result in discontinuous entities when the document is arranged in common reading order (*e.g.*, top-to-bottom and left-to-right).**

and then adapted to different document understanding tasks. These models have the capability to jointly encode the text, layout, and visual information of documents, enabling the generation of multi-modal representations that prove beneficial to downstream tasks.

Despite the above advancements made in improving document representations, the VrD-NER task still faces significant challenges due to the inherent characteristics of document data. It suffers from two common issues in real-world documents, complex document layouts and reading orders issues (see illustration in Figure-1). Previous studies solve these issues from two perspectives: (1) Introducing layout-related or spatial-aware pre-training tasks to enhance the ability of the models to understand the layout of documents [16, 24]; (2) Proposing datasets with annotations or pre-processing modules for reading order prediction to facilitate accurate identification of the order of token in documents [18, 29].

However, the current task formulation in VrD-NER hinders the progress made to solve the above issues. Many document transformers approach the VrD-NER task as a sequence labeling problem and utilize a sequence labeling (SL) head with the BIO-tagging scheme [19] for task-specific fine-tuning. Nevertheless, the SL head

is specifically designed for plain text in flat NER problems, thus cannot fully utilize multi-modal representations to learn reading order knowledge or represent discontinuous entities in documents. This significantly limits the application of these backbones and makes them fall short in handling the incorrect token order arrangement in real-world documents and discontinuous entities arising from complex layouts, leading to suboptimal performance in entity extraction and necessitating additional token serialization tools. To overcome this limitation, researchers have explored alternative task formulations. Zhang et al. [37] recognized the importance of reading order prediction in VrD-NER and proposed a token-path prediction head (TPP) to combine token order prediction and token classification in an integrated manner. However, the TPP head brings extra optimization targets and faces optimization issues. To obtain optimal performance, it requires substantial training samples for each entity type and performs poorly in low-resource settings.

Despite this, the SL head and the TPP head have a major drawback in that they are not effectively utilized for supervised pre-training, as they are designed for a fixed number of entity types. Recent studies have recognized the power of supervised pre-training in VrD-NER [23, 35]. Tang et al. [23] proposed to unify pre-training and multi-domain downstream tasks in a generative scheme and leveraged 11 supervised datasets for pre-training, which exhibits that adding supervised pre-training can further improve the model performance.

Based on the above observations, we recognize the significant benefits of developing a better prediction head for VrD-NER. To this end, we propose **UNER**, a **U**nified prediction head for **N**amed **E**ntity **R**ecognition. The UNER head is a query-aware entity extraction head that collaborates with existing document transformers to build better VrD-NER models. A UNER-based model can appropriately represent discontinuous entities and predict entities in correct reading orders. By harnessing the power of the supervised pre-training, it can effectively comprehend complex layouts and understand new documents with limited data.

Specifically, UNER models the VrD-NER task as a combination of sequence labeling task and reading order prediction, which is realized by two modules: (1) A query-aware token classification module that leverages the entity names (e.g., "address", "flight" and "units price" in Figure-1) as extraction clues for token-level classification; (2) A token order prediction module that predicts the pairwise token orders in the entities. By incorporating the predictions from the two modules, the UNER head is capable of learning the reading order knowledge from entity annotations and correctly predicting discontinuous entities in VrDs.

Additionally, since the UNER head can adaptively use any entity names as input queries, we can use a single UNER-based model to train on different NER datasets without the need for parameter reinitialization, thus enabling a supervised pre-training stage before fine-tuning. By leveraging supervised pre-training on external annotated documents with various layouts and entity types, we can incorporate universal layout understanding knowledge into the model and further enhance its performance on downstream VrD-NER tasks.

To assess the performance of UNER, we conducted experiments using 7 datasets from various domains and languages. The experimental results consistently demonstrated that UNER achieved significant improvements on all datasets, thereby showcasing its effectiveness as a prediction head for VrD-NER. Additionally, when applying a supervised pre-training stage, the UNER head exhibited notable advantages in cross-linguistic and few-shot settings, as well as demonstrating zero-shot VrD-NER capability. This discovery highlights the document knowledge transferability of UNER.

The contributions of our paper can be summarized as follows:

(1) We introduce UNER, a query-aware entity extraction head designed to address the challenges in VrD-NER, which works in collaboration with existing document transformers.

(2) The UNER head enables a supervised pre-training stage to enhance transformer backbones, which incorporates universal layout knowledge acquired from external pre-trained documents and results in improved performances in downstream VrD-NER tasks.

(3) Our experimental results consistently demonstrate that UNER can compete with previous methods on 7 benchmark datasets, and the incorporation of a supervised pre-training stage displays advantages in multiple settings.

## 2 RELATED WORK

**Pre-trained Document Transformers:** Recent research in pre-training techniques in NLP [2, 4, 39] and CV [1, 12] have demonstrated the significant potential in building multi-modal transformer for document understanding. Inspired by BERT [4], LayoutLM [30] improved the masked language modeling task to build a multi-modal document transformer that can effectively combine layout and textual information to understand and process documents with complex layouts. Following this idea, LayoutLMv2 [33] and LayoutLMv3 [6] focus on incorporating visual information and designing various cross-modal alignment pre-training tasks, and significantly enhance the model's performance in various document understanding tasks. Meanwhile, some studies have realized the importance of layout information and focus on improving text-layout interactions, such as StructuralLM [9], LiLT [26], LayoutMask [24], and GeoLayoutLM [16]. While the above models are pre-trained with self-supervised tasks, UDOP [23] proposed a supervised pre-training stage by unifying multiple VrDs tasks in a generative framework to fully exploit the correlation among different tasks. The UDOP model and universal NER studies in NLP [15, 34] have inspired us to propose a unified model for VrD-NER.

**Reading-order-aware Methods:** Understanding the reading order of documents remains a key challenge in VrD-NER and previous studies tried to address this issue from different perspectives. LayoutReader [29] proposed a sequence-to-sequence approach and a reading order dataset for order prediction. ERNIE-Layout [18] designed a reading order prediction task during pre-training and leveraged a serialization module to rearrange the order of input tokens. However, these methods require extra annotated data or introduce computation complexity. Inspired by discontinuous NER studies [10, 28], Zhang et al. introduced TPP, a prediction head to classify the document tokens and predict their correct reading orders in an integrated manner. These studies have inspired us to

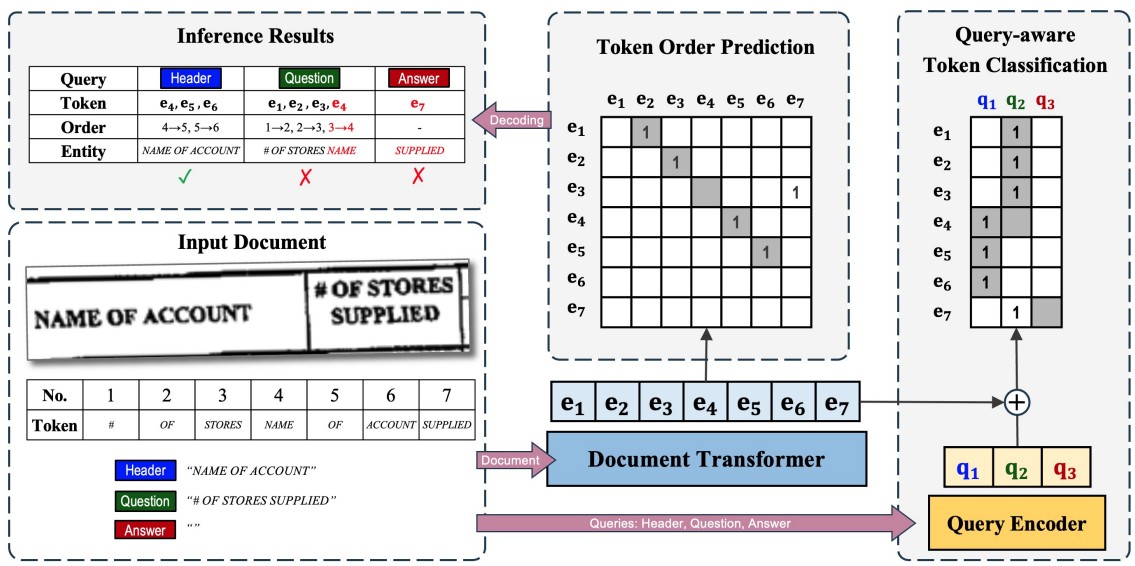

**Figure 2: An overview of the entity extraction pipeline for a document transformer using a UNER head. For better illustration, we use a document with a reading order issue as input. Given the input document, the UNER-based model receives the entity names ("header", "question", and "'answer") as queries and conducts token classification and order prediction in its two submodules, TOP and QTC. Here we use numbers to denote the ground-truth labels ("1" for positive and blanks for negative) and colored backgrounds to denote the binary classification predictions (gray for positive and white for negative). Ultimately, we combine the predictions for decoding and obtain the full entity spans. Incorrect predictions are denoted by the red color or a cross.**

reformulate the VrD-NER task and include a token order prediction task.

## 3 METHODOLOGY

The UNER head is an entity prediction head that works in collaboration with existing document transformers, which takes the token embeddings of the transformer backbone as input. Given an input document with $L$ tokens, a pre-trained document transformer is first utilized to obtain the token embeddings matrix $\mathbf{E} \in \mathcal{R}^{L \times h}$, where $h$ is the dimension of the embeddings. The $i$-th entry of $\mathbf{E}$, denoted as $\mathbf{e}_i \in \mathcal{R}^h$, represents the multi-modal embedding vector of the $i$-th token. These token embeddings are then inputted into the UNER head, which consists of two sub-modules: query-aware token classification and token order prediction. The pipeline of a UNER-based VrD-NER model is illustrated in Figure 2.

### 3.1 Query-aware Token Classification

Assuming that we need to extract $C$ types of entities from the document. when using a sequence-labeling-based (SL) head with the BIO-tagging scheme, the entity extraction task is treated as a $(2C + 1)$-way classification problem for each token embedding $\mathbf{e}_i$:

$$\mathcal{F}_{sl} : \mathbf{e}_i \rightarrow \{1, 2, ..., 2C + 1\}. \tag{1}$$

The function $\mathcal{F}_{sl}$ maps the token embedding $\mathbf{e}_i$ to a BIO-tagging label. However, this approach has a limitation in that it can only handle a fixed number of pre-defined entity types.

To address this issue, we propose the Query-aware Token Classification (QTC) module. This module first uses a query encoder to encode the entity names or descriptions as classification queries. It then leverages these queries as semantic clues to perform binary classification for all tokens.

To handle $C$ entity types, we start by encoding each entity name using a query encoder. This encoder transforms each entity name into a $h$-dimensional embedding, resulting in a query embedding matrix $\mathbf{Q} \in \mathcal{R}^{C \times h}$. In this matrix, each row corresponds to an entity type, and the $j$-th entry of $\mathbf{Q}$, denoted as $\mathbf{q}_j \in \mathcal{R}^h$, represents the embedding of the $j$-th query.

Then we combine the token embedding matrix and query embedding matrix to generate the query-aware token embedding tensor: $\mathbf{E} \oplus \mathbf{Q} \rightarrow \mathbf{S} \in \mathcal{R}^{L \times C \times h}$, where $\oplus$ stands for the broadcastable addition operation. We use $\mathbf{s}_{i,j}$ to denote $(i, j)$-th entry of $\mathbf{S}$, which is a query-aware token embedding for the $i$-th token and $j$-th query and satisfies: $\mathbf{s}_{i,j} = \mathbf{e}_i + \mathbf{q}_j$. For each $\mathbf{s}_{i,j}$, we will conduct a binary classification to determine if the $i$-th token belongs to the $j$-th entity type:

$$\mathcal{F}_{qtc} : \mathbf{s}_{i,j} \rightarrow \{0, 1\}. \tag{2}$$

We will conduct $L \times C$ binary classifications for all token-query pairs and the total classification loss is calculated as :

$$\mathcal{L}_{qtc} = -\frac{1}{LC} \sum_{i=1}^{L} \sum_{j=1}^{C} \text{CE}(\mathcal{F}_{qtc}(\mathbf{s}_{i,j}), \hat{s}_{i,j}). \tag{3}$$

Here $\text{CE}(\cdot, \cdot)$ is the cross-entropy loss function and $\hat{s}_{i,j} \in \{0, 1\}$ is the ground-truth label.

In QTC, the shape of the query-aware token embedding tensor is dynamically determined by the number of input queries. This flexibility allows us to input any number and type of queries into the module, thereby avoiding the limitation of pre-defined entity types in the SL head.

Moreover, as we use independent binary classifications, each token can be possibly classified into multiple categories. Such a design makes the UNER head capable of extracting nested and overlapped entities. While these entities are not fully represented in current VrD-NER benchmarks, they are common in real-world applications, as users may have different extraction goals within the same document and the same token may belong to multiple entity types. For instance, in Figure-1 case (1), a UNER-based model can simultaneously extract the full address (denoted in green), the city name ("Batang Kal"), and the state name ("Selangor") by using "address", "city", and "state" as input queries. In this case, these entities are overlapped and cannot be extracted with an exclusive $C$-way classification head.

Furthermore, the QTC module utilizes entity names as semantic clues for classification, which enables it to extract unseen entity types through semantic knowledge transfer.

### 3.2 Token Order Prediction

After token classification, it is necessary to arrange the tokens in order to form correct entity spans. In the SL head, the BIO-tagging scheme labels the "*Beginning*" and "*Inside*" information, which helps in token arrangement. However, it fails to handle discontinuous entities or tokens that are incorrectly ordered in VrDs. To overcome this limitation, we introduce a Token Order Prediction (TOP) module that predicts the token orders in VrDs.

The TOP module treats the token order prediction task as an edge prediction task on a token graph. For a document with $L$ tokens, we represent its token order using a directed graph with $L$ vertices. This graph is represented by a binary matrix $\mathbf{G} \in \{0, 1\}^{L \times L}$. If the $k$-th token is the successor of the $i$-th token, the $(i, k)$-th entry of $\mathbf{G}$, which represents the directed edge from $i$ to $k$, is set to 1. Otherwise, it is 0. By formulating the task in this way, the token order prediction becomes equivalent to predicting the edges on the graph.

During training, TOP performs binary classification for all directed edges in the graph:

$$\mathcal{F}_{top} : (\mathbf{e}_i, \mathbf{e}_k) \rightarrow \{0, 1\}. \tag{4}$$

The objective of this task is to optimize the following loss function:

$$\mathcal{L}_{top} = -\frac{1}{L^2} \sum_{i=1}^{L} \sum_{k=1}^{L} \text{CE}(\mathcal{F}_{top}(\mathbf{e}_i, \mathbf{e}_k), \hat{g}_{i,k}), \tag{5}$$

where $\hat{g}_{i,k} \in \{0, 1\}$ represents the ground-truth label for each edge in the graph.

It is important to note that $\hat{g}_{i,k}$ is derived from the entity annotations in the VrD-NER dataset, so it only covers the tokens that are part of entities. The order of the non-entity tokens is unknown during training, and therefore, the corresponding losses in Equation-5 should be ignored. Fortunately, after training with annotated documents, the knowledge of reading order learned from the entity-related tokens can be transferred to non-entity tokens

and tokens in other types of documents. Such knowledge transfer is discussed in detail in Section-4.4.

### 3.3 Optimization and Inference

The total training loss of the UNER head can be represented as follows:

$$\mathcal{L}_{total} = \mathcal{L}_{qtc} + \lambda\mathcal{L}_{top}, \tag{6}$$

where $\lambda$ is a hyper-parameter that controls the balance between the two losses. As shown in Figure-2, the ground truth labels in $\mathcal{L}_{qtc}$ and $\mathcal{L}_{qtc}$ are predominantly zeros, with only a small portion being ones. This label imbalance issue poses a challenge for loss optimization, so we employ the ZLPR loss [21] to enhance the training of the two losses and improve optimization efficiency.

During inference, we utilize the predictions of the QTC and TOP modules to construct full entity spans using the following steps:

(1) For each entity type (e.g., the $j$-th type), we identify all tokens that meet the condition $\mathcal{F}qtc(\mathbf{s}i, j) > 0$, along with their valid token orders satisfying $\mathcal{F}_{top}(\mathbf{e}_i, \mathbf{e}_k) > 0$.

(2) Based on the token orders, we identify the tokens without any predecessor tokens and use them as the starting tokens to create multiple token series.

(3) For each token series, we select the best subsequent token for the last token by choosing the token with the highest token order classification score and then adding it to the current token series.

(4) Repeat step (3) until the last token does not have any valid subsequent tokens, and use the resulting token series to form entity spans.

It should be noted that the UNER head enables multiple entity names to be queried for each document during both training and inference. When provided with a document and $C$ entity names, we only need to compute the document transformer and TOP module once and reuse their outputs to collaborate with the QTC module for $C$ iterations. This feature significantly alleviates the computational burden for recomputing the backbones when handling dense entity extraction tasks and enables a more flexible model deployment.

## 4 EXPERIMENTS

### 4.1 Datasets

In our experiments, we utilize 7 commonly used VrD-NER benchmarks to compare the performance of different methods. To evaluate the effectiveness of supervised pre-training in VrD-NER, we incorporate two additional datasets for pre-training, which encompass a wide range of document types and entity types. Detailed information of these datasets can be found in Table-1.

**Fine-tuning:** To demonstrate the effectiveness of our method, we perform fine-tuning on 7 popular VrD-NER benchmarks. Notably, when comparing with other prediction heads, we use the re-annotated versions of FUNSD [8] and CORD [17], namely FUNSD-r and CORD-r datasets [37]. The original datasets are not suitable for evaluating the ability to extract discontinuous entities, as the annotations in the original datasets are manually adjusted and do not reflect real-world scenarios. We use the re-annotated versions to better assess the entity extraction performance of different prediction heads. To provide solid experiments, we use another five

| Dataset | Language | Document Type | # of Samples (Train/Val/Test) | # of Entity Types | Stage |
|---------|----------|---------------|-------------------------------|-------------------|-------|
| FUNSD-r [37] | EN | Form | 149/ - /50 | 3 | FT |
| CORD-r [37] | EN | Form | 799/100/100 | 30 | FT |
| SROIE [7] | EN | Receipt | 626/ - /347 | 4 | FT |
| EPHOIE [27] | EN/ZH | Paper/Forms | 1183/ - /311 | 10 | FT |
| WildReceipt [22] | EN | Receipt | 1267/ - /472 | 25 | FT |
| SIBR [35] | EN/ZH | Invoice/Bill/Receipt | 600/ - /400 | 4 | FT |
| XFUND [32] | 7 Languages | Form | 1043/ - /350 | 4 | FT |
| SVRD [36] | ZH/EN | Various types | 1879/ - /1965 | 100+ | SP |
| DocILE [20] | EN | Invoice-like | 5180/100/500 | 55 | SP |

Table 1: The statistics of the datasets used in our experiments. "SP" and "FT" denote to be used during the supervised pre-training stage and fine-tuning stage, respectively.

| Backbone | Head | FUNSD-r | CORD-r |
|----------|------|---------|--------|
| LayoutLMv3 [6] | SL+BIO [19] | 78.77 | 82.72 |
| | TPP[37] | 80.40 | 91.85 |
| | UNER | **80.61** | **92.04** |
| LayoutMask [24] | SL+BIO [19] | 77.10 | 81.84 |
| | TPP [37] | 78.19 | 89.34 |
| | UNER | **78.90** | **89.95** |

Table 2: The F1 scores (%) of VrD-NER task on FUNSD-r and CORD-r datasets. We compare UNER with other prediction heads using two backbones, LayoutLMv3 and LayoutMask. The best results are denoted in boldface.

benchmarks when comparing UNER-based models with other baselines. These benchmarks, including a cross-linguistic dataset in 7 languages, cover a wide range of document types, entity types, and languages.

**Supervised Pre-training:** During supervised pre-training, we expect the dataset to have a better diversity, so we choose two large datasets, SVRD and DocILE [20] for pre-training. The SVRD dataset originates from the ICDAR 2023 Competition[1] and consists of densely annotated key-value linkings in various types of documents in real-world scenarios. By using the key entities as a description of the entity type, we transform these annotations into entity extraction labels, resulting in a VrD-NER dataset with hundreds of entity types. Since SVRD documents are primarily in Chinese, we utilize the DocILE dataset to enhance language diversity. This dataset comprises 5k documents with 55 entity types in English.

## 4.2 Experimental Settings

In our experiments, we compare different prediction heads using the base version of LayoutLMv3 and LayoutMask as transformer backbones, following the backbone setting in TPP. The query encoder in the QTC module is a 4-layer transformer with cross-attention to the transformer backbone. It is initialized with the first four

layers of Q-Former in BLIP-2 [11] with 38M parameters. We employ the tokenizer and the 1D-position embedding layer from the transformer backbone to process queries during query encoding. As each query can contain multiple tokens, we use the embedding of the first token as the query embedding. To determine the optimal value of the hyper-parameter $\lambda$, we conduct ablation studies. In our experiments, we set $\lambda = 0.1$ and $\lambda = 0.5$ for models with or without supervised pre-training, respectively. The batch size is set to be 16 for all fine-tuning datasets. For few-shot experiments, we repeat them five times and report the average scores. During the supervised pre-training stage, both the backbone and the UNER head are trained on the pre-training datasets for 20 epochs before fine-tuning.

## 4.3 Effectiveness of UNER

**Comparison with Prediction Heads:** We first conducted experiments to compare the performance of the UNER head with other prediction heads, namely the SL and TPP heads. Following the setting in TPP, we combined these heads with different backbones and evaluated their performance on two VrD-NER datasets, namely FUNSD-r and CORD-r. We use the entity-level F1 score as the evaluation metric. The experiment results are presented in Table-2, which demonstrate that the UNER head outperformed both the SL and TPP heads on both datasets, regardless of the backbone used. These results have demonstrated its superiority as a prediction head for the VrD-NER task.

It is important to highlight that the FUNSD-r and CORD-r datasets have discontinuous entities due to complex layouts and incorrectly ordered tokens. This significantly impacts the performance of the SL head, as it is unable to handle such entities. Additionally, the SL head is incapable of learning and conveying token order knowledge effectively, which limits its usefulness as a VrD-NER head.

While the TPP head is capable of handling discontinuous entities, it faces optimization issues that negatively affect its performance. This is because the TPP head attempts to jointly optimize token classification and token order prediction in a unified manner. For each entity, it predicts one token graph to represent entity-specific token paths, resulting in the need to predict a $L \times L \times C$ binary tensor and $L^2C$ classification sub-tasks. Such entity-specific token

---

[1]https://rrc.cvc.uab.es/?ch=21&com=introduction

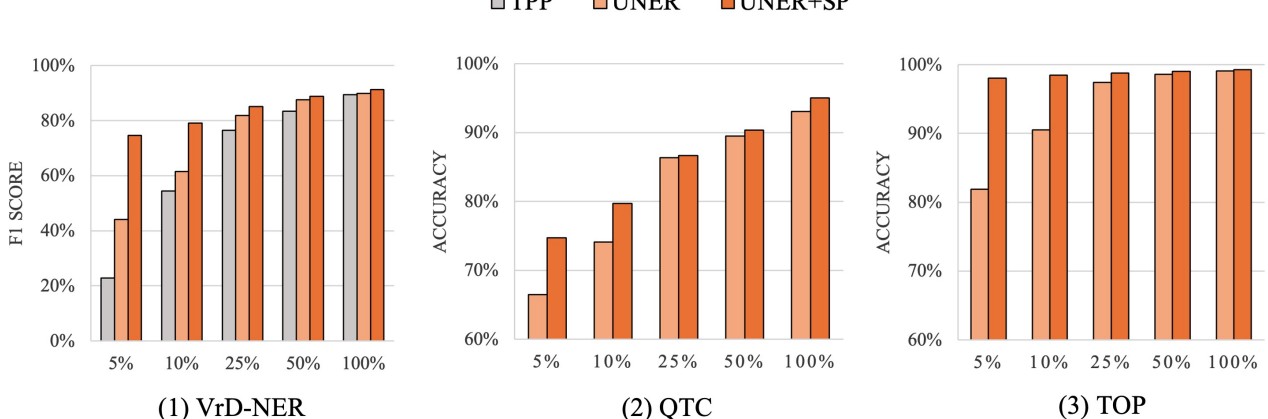

(1) VrD-NER

(2) QTC

(3) TOP

**Figure 3: The few-shot performance on the CORD-r dataset when using different percentages of training samples (from 5% to 100%). We use LayoutMask as the backbone and compare its performance with different prediction heads or supervise pre-training conditions. (1): The entity-level F1 scores in VrD-NER when using TPP, UNER, and UNER with supervised pre-training ("UNER+SP"). (2)&(3): For the UNER-based method, we also report the performance of its submodules, the token-level classification accuracy in QTC, and the token order classification accuracy in TOP. As we do not have the complete reading order annotations for all the tokens in the documents, in TOP we only calculate the accuracy for entity-related tokens.**

graphs in TPP also influence the transfer of token order knowledge among different entity types.

On the other hand, the UNER head has two separate optimization goals and reduces the number of binary classifications sub-tasks to $(L^2+LC)$. Additionally, UNER explicitly learns and represents token order knowledge in one unified token graph which is independent of the entity types, making it suited for transferring universal token order knowledge. This design makes the UNER head easier to optimize and more efficient with limited data.

To further highlight this advantage, we compared the TPP and UNER head on the CORD-r dataset in a few-shot setting, where the model is fine-tuned using only a subset of the training samples. The results of this comparison are summarized in Figure-3. The findings from these experiments highlight that UNER consistently outperforms TPP across all training percentages. It is noteworthy that the performance advantage of UNER becomes more pronounced as the percentage of training data decreases. For instance, when utilizing only 5% of the data, UNER achieves a notable improvement of +21.24% compared to TPP. This demonstrates the data efficiency of UNER in handling limited training data.

**Comparison with VrD-NER Baselines:** The above results have proved the advantage of UNER compared to other prediction heads. In this section, in order to show that the UNER head can help in building excellent VrD-NER models, we further extended our comparison of two UNER-based models against other *state-of-the-art* VrD-NER models on 5 datasets: SROIE, EPHOIE, WildReceipt, SIBR, and XFUND.

The results of these comparisons are presented in Table-3. Note that XFUND is a cross-linguistic dataset and comprises seven sub-datasets, so we report the averaged F1 scores across them. The UNER head demonstrates highly competitive results on all datasets with both backbones, and it achieves the highest F1 score on three

out of the five datasets overall, which is a significant achievement considering the number and diversity of the compared benchmarks. Notably, "LayoutLMv3+UNER" outperforms other methods by a significant margin (+2.93%) on the WildReceipt dataset. These results further validate the effectiveness and compatibility of our UNER head to work with document transformer backbones.

**Comparison in Multi-task Setting:** In order to demonstrate the potential of UNER in unified entity extraction, we conducted a multi-task experiment using four datasets: SROIE, EPHOIE, WildReceipt, and SIBR. The main objective was to illustrate that the UNER head can effectively handle multiple datasets with one model. We accomplished this by fine-tuning the UNER-based model on the combined training samples from these four datasets and evaluating the model's performance separately. The results are presented in Table-3 as "+*Multi-task*". The UNER-based model in the multi-task setting achieved competitive results compared to other methods, highlighting its potential as a unified VrD-NER head.

However, we also observed that the F1 scores of the UNER-based models in the multi-task setting were slightly lower compared to the scores in a single-task setting. We believe that this is due to the difficulty in balancing the number of training samples for different datasets. As our focus is not on searching for the best performance with a particular group of datasets, we simply combined these datasets for training without making careful adjustments in order to showcase the vanilla capability of UNER for unified VrD-NER.

## 4.4 Effectiveness of Supervised Pre-training

The UNER head has a significant advantage as it can be utilized for supervised pre-training, which further enhances the self-supervised

| Method | SROIE | EPHOIE | WildReceipt | SIBR | XFUND |
|---|---|---|---|---|---|
| GAT [25] | 87.23 | 96.90 | 85.43 | - | - |
| RoBERTA [14] | - | 95.21 | - | - | 71.00 |
| TRIE [38] | 96.18 | 93.21 | 85.99 | 85.62 | - |
| SDMG-R [22] | 87.10 | - | 88.70 | - | - |
| LayoutXLM [31] | - | 97.59 | - | 94.72 | 80.72 |
| StrucTexT [13] | 96.88 | 97.95 | - | - | - |
| LiLT [26] | - | 97.97 | - | - | 82.28 |
| TRIE++ [3] | 96.80 | _98.85_ | 90.15 | - | - |
| ESP [35] | - | - | - | 95.27 | **87.27** |
| GeoLayoutLM [16] | **97.97** | - | - | - | - |
| LayoutLMv3 [6]+UNER | 97.39 | 98.71 | **93.08** | 95.70 | 82.66 |
| *+Multi-task* | *96.35* | *98.39* | *92.45* | *95.13* | *-* |
| LayoutMask [24] +UNER | _97.61_ | **99.10** | _92.53_ | _95.40_ | _83.87_ |
| *+Multi-task* | *97.36* | *97.58* | *92.00* | *94.75* | *-* |

**Table 3: The F1 scores (%) of the VrD-NER task on SROIE, EPHOIE, WildReceipt, SIBR, and XFUND. The best results for each dataset are denoted in boldface and the second best results are underlined. The results in a multi-task setting are in grey color.**

| Method | ZH | JA | ES | FR | IT | DE | PT | Average |
|---|---|---|---|---|---|---|---|---|
| RoBERTa [14] | 87.74 | 77.61 | 61.05 | 67.43 | 66.87 | 68.14 | 68.18 | 71.00 |
| LayoutXLM [31] | 89.24 | 79.21 | 75.50 | 79.02 | 80.82 | 82.22 | 79.03 | 80.72 |
| XYLayoutLM [5] | _91.76_ | 80.57 | 76.87 | 79.97 | 81.75 | 83.35 | 80.01 | 82.04 |
| LiLT [26] | 89.38 | 79.64 | 79.11 | 79.53 | 83.76 | 82.31 | 82.20 | 82.28 |
| ESP [35] | 90.30 | _81.10_ | **85.40** | _90.50_ | **88.90** | _87.20_ | **87.50** | **87.27** |
| LayoutMask [24] +UNER | 91.43 | 79.28 | 78.23 | 83.57 | 82.84 | 86.40 | 85.37 | 83.87 |
| LayoutMask [24] +UNER+SP | **91.83** | **81.40** | _82.77_ | **90.54** | _87.33_ | **88.59** | _85.86_ | _86.90_ |

**Table 4: The F1 scores (%) of the VrD-NER task on the sub-datasets in XFUND. "+SP" denotes the utilization of a supervised pre-training stage. The best results are denoted in boldface and the second best results are underlined.**

pre-trained backbones. To assess such potential, we leverage two annotated VrD-NER datasets, SVRD and DocILE, to incorporate supervised pre-training before fine-tuning. We use "UNER+LayoutMask" as the base model and compare its fine-tuning performance with and without supervised pre-training. We find that the inclusion of a supervised pre-training stage improves models with notable effectiveness in cross-linguistic and few-shot scenarios and brings zero-shot VrD-NER abilities.

**Comparison in Cross-linguistic Setting:** Table-4 presents the detailed F1 scores of the sub-datasets in XFUND, a multi-linguistic dataset comprising 7 languages. The disparity in results between "LayoutMask+UNER" and "LayoutMask+UNER+SP" configurations highlight the efficacy of the supervised pre-training stage in enhancing the UNER-based model across all languages, which results in an average increase of 3.03% in F1 score. Moreover, when compared to other methods, "LayoutMask+UNER+SP" achieves the highest performance on four sub-datasets: ZH, JA, FR, and DE. It is worth mentioning that the backbone model is pre-trained on English documents while the supervised pre-training datasets only include English and Chinese documents, so six out of the seven languages

in XFUND are unseen before fine-tuning, which proves the effectiveness of supervised pre-training for cross-linguistic VrD-NER tasks.

**Comparison in Few-shot Setting:** In this setting, we compare the performance of the UNER-based model with and without supervised pre-training after fined-tuned with a subset of the training samples. As the performance of entity extraction in UNER relies on the accuracy of its sub-modules, we also calculate the accuracy scores of token classification and token order prediction in the QTC and TOP modules to gain a more comprehensive understanding. The results are listed in Figure-3.

Our observations indicate that supervised pre-training has significant benefits in few-shot scenarios. Specifically, the "UNER+SP" model achieves remarkably high performance even with a small portion of training samples, outperforming the "UNER" model by a large margin on all tasks and configurations. Notably, the f1 score of "UNER+SP" in "VrD-NER" is surprisingly high with extremely limited data (74.62% with only 5% training samples). Furthermore, "UNER+SP" with 5% training samples has a higher accuracy score in "TOP" than "UNER" with 25% samples (98.01% compared to 97.39%). These advantages highlight that the UNER-based model benefits

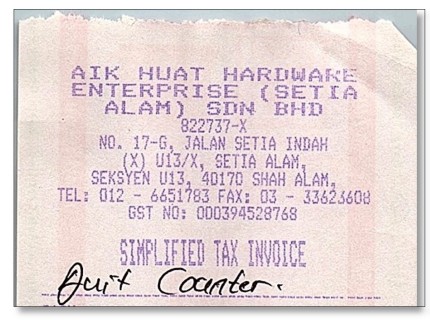 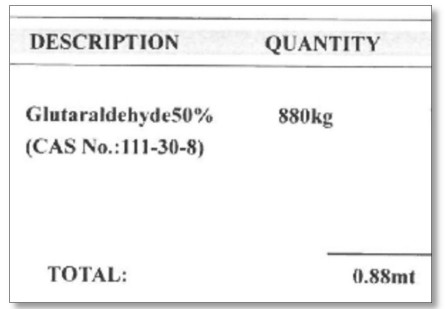

| Query | Prediction | Confidence |
|---|---|---|
| Header | AIK HUAT HARDWARE | 0.86 |
| TEL | 012-6651783 | 0.98 |
| 电话 *(Telephone)* | 012-6651783 | 0.95 |
| FAX | None | - |

**(1)**

| Query | Prediction | Confidence |
|---|---|---|
| DESCRIPTION | Glutaraldehyde50% (CAS No.: 111-30-8) | 0.95 |
| CAS No | :111-30-8 | 0.82 |
| QUANTITY | 880kg | 0.99 |

**(2)**

| Query | Prediction | Confidence |
|---|---|---|
| 座位号 *(Seat Number)* | 7A | 0.99 |
| Seat | 7A | 0.98 |
| 登机时间 *(Boarding Time)* | 0850 | 0.94 |
| Boarding Time | 0850 | 0.94 |

**(3)**

**Figure 4: Visualization of the entity predictions. We pre-train "LayoutMask+UNER" with SVRD and DocILE and display the predicted entities and their confidence scores with various queries. The grey texts in the brackets serve as translations and are not used as input queries. Incorrect predictions are highlighted in red. Incorrect predictions are denoted by red color. (1): A receipt from SROIE with queries that differ from the entity types in the original dataset. (2): Extraction of overlapped entities with vertical alignment in SIBR. (3): A bilingual air ticket with a misaligned layout.**

| SP Settings | | SROIE | EPHOIE |
|---|---|---|---|
| **SVRD** | **DocILE** | *QTC/TOP/VRD-NER* | *QTC/TOP/VRD-NER* |
| √ | | 15.14 / 48.30 /  0.07 | 57.38 / 93.77 / 27.14 |
| | √ | 0.00 / 75.91 /  0.00 | 0.00 / 83.81 /  0.00 |
| √ | √ | 18.70 / 99.44 / 21.90 | 71.65 / 96.56 / 36.86 |

**Table 5: The zero-shot prediction results of the UNER-based model after supervised pre-training. We pre-train the "LayoutMask+UNER" with varying supervised pre-training datasets and assess the zero-shot performance on SROIE and EPHOIE. To offer a comprehensive insight, we present QTC accuracy, TOP accuracy, and F1 score in VrD-NER.**

significantly from the document knowledge learned during pre-training on large datasets. This knowledge facilitates the model's adaptation to the other downstream VrD-NER dataset, particularly when the available data is limited.

**Comparison in Zero-shot Setting:** In order to further explore the effectiveness of knowledge transfer through supervised pre-training, we conducted pre-training of the UNER-based model using various combinations of datasets and assessed their zero-shot performance on other VrD-NER benchmarks. The pre-trained models exhibited the ability to extract entities in a zero-shot setting on the SROIE and EPHOIE datasets, and the results are summarized in Table-5. In addition to overall VrD-NER performance, we also provide zero-shot accuracy scores for the QTC and TOP sub-modules for a more comprehensive analysis.

Our analysis indicates that the zero-shot entity extraction ability is largely attributed to the inclusion of the SVRD dataset, which significantly improves the QTC for both SROIE (15.14%) and EPHOIE

(57.38%). Furthermore, incorporating the DocILE dataset further enhances the VrD-NER abilities, particularly in SROIE where the F1 score increased from 0.07% to 21.90%.

In order to gain a better understanding of the entity extraction ability of the pre-trained model, we tested the best model (pre-trained on both SVRD and DocILE) with various documents and queries, and presented some prediction results in Figure-4. Our observations indicate that the model is effective in extracting entities with explicit trigger words. Using these trigger words as queries consistently yields satisfactory results, regardless of the layout alignment. However, the model exhibits limitations when handling entities without explicit trigger words, unnatural queries, or queries with varying meanings. For example, the entity names in the CORD-r dataset are specifically designed, such as "menu.sub.price" and "total.cashprice". Similarly, the entity type "Header" is used in multiple datasets, but its meaning can differ across documents, leading to inconsistent predictions. These findings underscore the potential for using diverse pre-training datasets to develop a unified VrD-NER model with stronger zero-shot abilities, while also identifying challenges in transferring document understanding knowledge.

## 5 CONCLUSION

This paper presents UNER, a query-aware entity extraction head for unified named entity extraction to address the challenges in VrD-NER. Experimental results across various datasets demonstrate the superior performance of UNER compared to existing methods. Moreover, UNER exhibits knowledge transferability when supervised pre-training is applied and contributes to building robust backbones for VrD-NER tasks.

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
