# OpenReview forum: "UNER: A Unified Prediction Head for Named Entity Recognition in Visually-rich Documents"
_acmmm.org/ACMMM/2024/Conference — MM2024 Poster_

### Official Review · Reviewer_WJfQ · 2024-05-25

**Rating:** 3
**Confidence:** 2

**Summary:**

The recognition of named entities in visually-rich documents (VrD-NER) is essential but challenging due to complex layouts, incorrect reading orders, and unsuitable task formulations. To tackle these challenges, the proposed query-aware entity extraction head, UNER, collaborates with multi-modal document transformers to improve VrD-NER models. UNER addresses issues of discontinuous entities by combining sequence labeling and reading order prediction, demonstrating effectiveness through experiments on diverse datasets. Additionally, pre-trained UNER models show significant advantages in few-shot, cross-linguistic, and zero-shot entity extraction scenarios.

**Strengths:**

1. The experiments were comprehensive.
2. Experimental results across various datasets demonstrate the superior performance of UNER compared to existing methods.

**Limitations:**

1. The authors did not provide source code for reproducibility.
2. How should the term (2C+1)-way in line 283 be understood? Please provide a more detailed explanation.
3. It would be better to use italics for mathematical symbols in the manuscript, such as the matrix \textbf{E} in line 270.
4. Please explain the Multi-task experiment mentioned in Table 3.
5. In the experiments, documents are used as model input; however, from the sample shown in the figure, it can be seen that the documents are in image format. How is this handled?
6. During fine-tuning, did the authors fine-tune using all the data from the seven datasets together?
7.  There are significant gaps between paragraphs in the article that need to be adjusted appropriately.

**Suitability:**

2

---

### Official Review · Reviewer_pY9U · 2024-05-25

**Rating:** 3
**Confidence:** 2

**Summary:**

The paper introduces UNER, a unified prediction head for Named Entity Recognition in visually-rich documents. UNER models VrD-NER as a combination of sequence labeling and reading order prediction, addressing complex layouts and discontinuous entities. It enables supervised pre-training on various datasets, incorporating universal layout understanding into document transformer backbones, leading to improved performance and cross-lingual capabilities.

**Strengths:**

1. UNER addresses the common issues of complex layouts and incorrect reading orders in VrD-NER, enabling the representation of discontinuous entities and the prediction of entities in correct reading orders.
2. The incorporation of a supervised pre-training stage using UNER enhances the document transformer backbones with universal layout understanding, leading to improved performance in downstream VrD-NER tasks.

**Limitations:**

1. The experimental results seem to show only modest improvements, and the paper does not demonstrate significant advantages of the proposed method over existing approaches.
2. The query-aware extraction approach may require specifying entity types multiple times, potentially impacting performance to some extent.
3. It would be interesting to analyze whether visual-related NER tasks can be effectively handled by current multi-modal language models, such as GPT-4V, and how these models perform on such tasks compared to UNER-based models.

**Suitability:**

2

---

### Official Review · Reviewer_UqH9 · 2024-05-27

**Rating:** 5
**Confidence:** 1

**Summary:**

This work aims to deal with the visually-rich name entity recognition problem and tackle some critical issues, such as the diverse layout information and incorrect reading order. To solve these problems, they propose a Unified prediction head, a query-aware entity extraction head that collaborates with both the token representation and read order information. They conduct experiments under various domains, language, and settings.

**Strengths:**

1. The topic is quite interesting and significant for the research field. The author tackled quite important issues in this area and conducted adequate experiments to prove the effectiveness of their methods.
2. This method achieves better performance in the zero-shot learning setting and shows good potential in language transferability.

**Limitations:**

1. Some things are not clear, maybe I am not familiar with the field.  For example, the entity type in the given example in Figure 2, what is the 'Answer' label mean?

**Suitability:**

3

---

### Official Review · Reviewer_ZsqC · 2024-05-30

**Rating:** 4
**Confidence:** 3

**Summary:**

The paper addresses challenges in Named Entity Recognition for Visually-rich Documents (VrD-NER) by proposing UNER, a query-aware entity extraction head. UNER works with existing document transformers to improve VrD-NER models by combining sequence labeling and reading order prediction. This approach effectively handles complex layouts and discontinuous entities. Additionally, UNER supports supervised pre-training, enhancing model performance on downstream tasks.

**Strengths:**

- The proposed UNER is novel, addressing the issues of complex document layouts and reading order, and is orthogonal to previous methods.
- The authors conducted thorough experimental analysis to demonstrate the effectiveness of UNER.
- UNER shows excellent generalization, performing well on unseen languages.

**Limitations:**

- The authors need to briefly introduce the methods used in the experimental section within the related work chapter.
- In Table 3, there are many missing values. The authors should explain why these values were not reported.
- The authors should provide an explanation for why the ESP model performs better than UNER in some multilingual domains.

**Suitability:**

3

---

### Meta-Review · Area_Chair_Hz28 · 2024-06-28

**Recommendation:** Accept (Poster)
**Confidence:** 5

**Metareview:**

The paper presents a novel approach, UNER, a query-aware entity extraction head designed to enhance Named Entity Recognition for Visually-rich Documents (VrD-NER). By combining sequence labeling with reading order prediction, UNER effectively handles complex document layouts and discontinuous entities. The method is shown to generalize well, performing robustly in zero-shot and cross-lingual settings. Reviewers praised the comprehensive experimental analysis and the method's novelty in addressing challenging issues in VrD-NER. However, some concerns were noted, including the need for more detailed explanations of certain experimental methods and the modest improvements observed in results. Despite these minor issues, the paper's contributions are significant, demonstrating clear advancements in VrD-NER and justifying acceptance.